# Insights into Environmental Drivers on the Reproductive Cycle of *Diopatra neapolitana* (Polychaeta: Onuphidae)

**DOI:** 10.3390/biology11101504

**Published:** 2022-10-14

**Authors:** Daniel Escobar-Ortega, Lucía Couceiro, Ramón Muíño, Edgar No, Nuria Fernández

**Affiliations:** 1Facultade de Ciencias, Universidade da Coruña, Campus de A Coruña, 15071 A Coruña, Spain; 2Centro de Investigacións Mariñas, Consellería do Mar, Pedras do Corón s/n, 36620 Vilanova de Arousa, Spain

**Keywords:** taxonomy, bristle worm, reproduction, Galicia, maturation cycle

## Abstract

**Simple Summary:**

The environmental variables that drive the reproductive cycle of *Diopatra neapolitana* were studied in a population of Ría de Vigo (NW Spain). The identity of the species was corroborated using the six diagnostic morphological characters. A discontinuous reproductive cycle was observed, with a period of proliferation and oocyte maturation from January to September, asynchronous spawning events from July to October and a resting period in November and December. The reproductive cycle seems to be conditioned by the photoperiod and the water temperature. In particular, the sudden drop in water temperature, as a result of coastal upwelling processes that typically occur in the summer months of the study area, was found to induce spawning in this species.

**Abstract:**

The reproductive cycle of *Diopatra neapolitana* was studied over two years in Redondela (NW Spain), considering both the monthly evolution of the percentage of individuals with oocytes and the variation of the mean oocyte size. Moreover, since three other species of *Diopatra* have recently been documented in regions geographically close to NW Spain, we first confirmed the identity of this species by carrying out a morphological identification of a representative number of the specimens collected. Our data showed a discontinuous reproductive season, with a period of proliferation and maturation of oocytes from January to September and asynchronous spawning events between July and October. Then, a resting period in the months of November and December was observed. We also explored the influence of some relevant environmental factors (namely, water temperature, photoperiod, salinity, primary production and upwelling index) on the observed reproductive cycle. The results suggest that water temperature is one of the most important drivers of the reproductive cycle for this species. Particularly, in Ría de Vigo, the reproductive cycle seems to be influenced by upwelling events of deep cold water that occur frequently between March and September, and that could act as a spawning-inducing switch.

## 1. Introduction

The genus *Diopatra* includes around 50 tube-dwelling polychaete annelid species in the family Onuphidae. They are characterized by building long, emergent tubes that play an important role in the marine sediment. Among others, dense clumps of tubes act as local predation refugia for the infauna [1], stabilize the sediment [2] and catch sediments and drift algae that provide a nutritionally rich system for other organisms [3], facilitating the appearance of other species [4], such as some macroalgae [5]. In addition, some birds and fish feed directly on these polychaetes [6].

Many *Diopatra* species are common in tropical and temperate regions [7]. In particular, *Diopatra neapolitana* occurs in intertidal and shallow subtidal depths along the Atlantic Iberian and French coasts and throughout the Mediterranean [8], occasionally reaching densities as high as 1500 individuals/m^2^ [9]. Records from elsewhere probably relate to different but morphologically similar species [10], although in recent years it was reported in Araçá Bay, Brazil, as an introduced species [11]. Until recently, *D. neapolitana* was the only species of the genus described along the European coasts. However, this list has grown in the last decade and up to three more *Diopatra* species have currently been documented on the west coast of Europe. Specifically, *Diopatra marocensis* and *Diopatra micrura* have been recorded on the continental coast of Portugal [7,12,13], while *D. marocensis* and *Diopatra biscayensis* have been registered on the western coast of France [8,13], the latter also being observed more recently in the Cantabrian Coast [14]. However, in Galicia (NW Spain), only the existence of *D. neapolitana* has been documented up to date.

Galician populations of *D. neapolitana* represent an important marine resource that is exploited for use as recreational fishing bait, as it occurs in other countries [15,16,17,18]. In this context, the stock assessment of exploited populations has traditionally incorporated reproductive data in the pursuit of sustainable exploitation, such as length at first maturity and fecundity [19]. Moreover, the role that reproductive biology plays in the population productivity and the need to understand factors driving this productivity [20,21,22] is being increasingly recognized as an essential component of effective fisheries management [23]. This is evidenced by a large number of recent papers that highlight the relationship between reproductive biology and stock sustainability [24,25,26]. Beyond that, knowledge of environmental drivers of the reproductive biology of species is increasingly required in the context of climate change to predict how alterations in the reproduction patterns related to temperature may affect the distribution of the species [27,28].

The scientific literature shows considerable differences in the reproductive biology within the genus *Diopatra.* Indeed, Paxton (1993) [29] distinguished four groups of *Diopatra* species according to their developmental pattern. These four groups include direct development with brooding inside the parental tube as it occurs, for example, in *Diopatra marocensis* [30,31,32] and *Diopatra nishii* [33]; direct development with egg masses attached to the parental tube, as in the case of *Diopatra maculata* [29]; early trochophore development in egg masses attached to the parental tube and posterior release of lecithotrophic larvae after jelly mass disintegration (e.g., *Diopatra biscayensis)* [13,14]; or broadcast spawning with a brief pelagic larval stage, as in *D. neapolitana* [10,11,16,34]. More specifically, the studies focusing on the reproduction of *D. neapolitana* show that the oogenesis is extraovarian, with previtellogenesis and vitellogenesis occurring free in the coelom [10,11]. Immature oocytes exhibit two strings of nurse cells attached to them, which probably transport nutrients from the coelom fluid to the developing oocytes [10,11,16,34]. It has been found as well that the fertilisation occurs in the water column. After that, the embryos transform in a free-swimming lecithotrophic larvae that begin to develop as part of the plankton community. After 3–4 days, the larvae transform into small juveniles with up to five differentiated segments [9,34,35] that settle in the sediment [34]. Other interesting observations have recently been made, such as the discovery that *D. neapolitana* is a protandrous hermaphroditic species, characterized by the presence of spermaducal papillae [10,11], and not dioecious as previously believed [16,34]. Moreover, the reproductive cycle has been described as continuous throughout the year in Izmir Bay, eastern Mediterranean [16] and Araçá Bay, SW Atlantic [11], or as taking place at a specific period of the year in Ria de Aveiro, NW Portugal [36] and the Villaviciosa estuary, northern Spain [10].

Although it has been shown that some environmental factors can affect the biology of *Diopatra*, such as its regenerative capacity [37], there are scarce studies on the influence of these environmental factors on its reproductive biology, apart from a couple of studies that show the importance of temperature in larval development [34], and in the possible stimulation of gametogenesis and spawning [30].

In order to develop the knowledge of the reproductive biology of *Diopatra neapolitana*, the reproductive cycle of this species, as well as its relationship with some relevant environmental variables, has been studied in Ría de Vigo (Galicia, NW Spain). Additionally, due to the recent records of other *Diopatra* species in regions geographically close to Galicia, and the differences in the reproductive biology within this genus, the taxonomic identity of the sampled species was confirmed.

## 2. Material and Methods

### 2.1. Study Area and Sampling

*Diopatra* specimens were collected in the shellfish bank A Portela (VI-132), within Ría de Vigo (Redondela, NW Spain) (Figure 1). This is a very heterogeneous intertidal zone that includes muddy, sandy and rubble areas. Here, *Diopatra* sp. is distributed and can be collected all over the bank, regardless of the granulometry or the habitat.

Specimens were provided by a professional bristle worm fisher at monthly intervals from May 2019 to April 2021. About 40 individuals (from May 2019 to September 2019) and about 60 individuals (from October 2019 to April 2021) were provided each month, with the exception of April 2020 and May 2020, when restrictions arising from COVID-19 prevented this quantity from being obtained. Individuals were captured by pouring salt inside the tube wherein the animal lived. This forced the animal to come out, at which moment it had to be swiftly captured.

### 2.2. Treatment of Live Specimens

The collected specimens were transported alive in sea water to the laboratory the same day of their capture. Once there, the individuals were weighed after being dried with blotting paper. We also measured the width of the 10th chaetiger (W_10_), using ToupView software for image analysis. The W_10_ is commonly used as a proxy of body size in *Diopatra* studies [10,11,12,16], due to the fact that many specimens are sectioned at the capture time, or in response to stress during the transport or handling. Finally, the whole animals were individually preserved in 70% ethanol.

### 2.3. Morphological Identification

The taxonomic identity of the *Diopatra* specimens was verified following Pires et al. (2010) [11], who proposed the study of several morphological characters. In their study, these authors included *D. biscayensis* specimens from France and *D. micrura* specimens from Ría de Aveiro; in addition, they also added to this dataset *D. neapolitana* and *D. marocensis* specimens from Ría de Aveiro obtained from Rodrigues et al. (2019) [12]. This study demonstrates that six morphological characters accurately discriminate these four *Diopatra* species.

We studied these six morphological characters (width of the 10th chaetiger (W_10_), number of rings in the ceratophores, maximum number of branchiae whorls, first chaetiger with subacicular hooks, number of teeth in the pectinate chaetae and presence/absence of a ventral lobe in the parapodia) on 10% of the total sample (105 randomly selected individuals). Detailed scanning electron microscope images of the morphological characters studied can be found in Appendix A of the Appendix A. The obtained data were incorporated into a representative subsample of the specimens from the Pires et al. (2010) [11] database that was provided to us by its authors and analysed as described in Section 2.5 to determine the species present in the study area.

### 2.4. Reproductive Cycle

The presence of oocytes in the coelom was checked in all fixed specimens, after dissection of the body region between the 30 and the 50 post-branchial setigers, by making an incision with a scalpel. The mean oocyte size was estimated from an aliquot of coelomic solution extracted with a Pasteur pipette. This coelomic solution was first filtered through a 60 µm sieve to discard tiny coelomic debris and the particles retained were next filtered through a 500 µm sieve. The eluted solution was collected, mounted in a Petri dish and examined under a stereomicroscope with a digital camera attached. The NIS-Elements image analysis software (Nikon Instruments Inc., Melville, NY, USA) was then used to determine the diameter of 45 oocytes picked at random. Therefore, for the purposes of this study, ovigerous females were those females with oocytes between 60 and 500 µm. Still, we also qualitatively recorded the presence of oocytes smaller than 60 µm by checking the solution that was not retained in the 60 µm sieve under a stereomicroscope.

We also explored the influence of some relevant environmental variables on the reproductive cycle of this species. In particular, water temperature, salinity and fluorescence and UV fluorescence as indicators of primary production, were retrieved from the Rande oceanographic station belonging to INTECMAR (http://www.intecmar.gal/; accessed on 11 June 2022), using the available data at 3 m of depth closest to our sampling dates. Photoperiod in the sampling dates was obtained from the MeteoGalicia historical database for Vilagarcía de Arousa (https://www.meteogalicia.gal/; accessed on 11 June 2022). Finally, upwelling index time series were provided by the Instituto Español de Oceanografía (www.indicedeafloramiento.ieo.es; accessed on 11 June 2022).

### 2.5. Statistical Analyses

Morphological data in relation to the six diagnostic characters mentioned in Section 2.3 and obtained from individuals sampled in the study area were added to a subsample of the data matrix constructed by Pires et al. (2010) [11]. This data matrix comprised 51 Diopatra individuals belonging to 4 different species: six individuals of *D. biscayensis* (five from Arcachon Bay, France, and one from Marennes Oléron, France), 15 individuals of *D. neapolitana*, 15 of *D. marocensis* and 15 of *D. micrura*, all of them from Ria de Aveiro, Portugal. Following normalisation of the variables, the morphological data matrix was submitted to classification, using Unweighted Pair Group Mean Average upon the Euclidean distance matrix between the specimens, and ordination, using principal components analysis, with the software PRIMER-e [38].

The monthly variation of the mean oocyte size was studied. In addition, main spawning events were identified by sharp drops in the percentage of individuals with oocytes.

We used correlations to explore the relationship between the reproductive descriptors (i.e., mean oocyte size and percentage of individuals with oocytes > 60 µm) and the environmental variables. Normality of the variables was checked by Shapiro–Wilk test. Pearson’s correlation test was applied to normally distributed variables while non-parametric Kendall rank correlation test was run with non-normal variables. We also explored the correlation between the mean oocyte size and the W_10_, as well as between the mean oocyte size and the weight of the individuals. The correlations were calculated using IBM SPSS Statistics for Windows, Version 25.0 (IBM Corp. Released 2017, Armonk, NY, USA: IBM Corp).

## 3. Results

A total of 1130 specimens were collected from May 2019 to April 2021. No samples were obtained in December 2019 and January 2021 due to weather difficulties, and in April 2020 due to the COVID-19 mobility restrictions (Table 1).

### 3.1. Morphological Identification

Table 2 summarizes the values of the key diagnostic morphological characters employed for the taxonomic identification of the *Diopatra* specimens collected in the present work. As it can be observed in this table, these values were similar to those studied by Pires et al. (2010) [11] for *D. neapolitana* specimens collected in Ria de Aveiro (NW Portugal). The principal component analysis (PCA) (Figure 2) showed four well isolated clouds of points: one formed by the 15 individuals of *D. micrura*; another formed by the 15 individuals of *D. marocensis*, a third one comprising the six individuals of *D. biscayensis* and, finally, a fourth group encompassing both the 15 individuals of *D. neapolitana* originally included in the data matrix of Pires et al. (2010) and the 105 individuals studied in the present work. The PCA axis 1 and 2 comprised 86.51% of the total variance. The individuals collected in Redondela were thus well isolated from *D. marocensis* and *D. biscayensis*, as these two species were grouped at the opposite end of the ordination axis 1. *D. micrura* specimens occupied an intermediate position between the *D. marocensis* and the *D. biscayensis* clusters and the cluster formed by *D. neapolitana* as well as our individuals; still, these *D. micrura* specimens were isolated from the other three species in the positive pole of the ordination axis 2.

### 3.2. Reproductive Cycle

A total of 300 out of the 1130 specimens analysed (26.5%) contained free oocytes in their coelom. We did not observe differences in body colour between ovigerous females and all other individuals (whether male or immature). Indeed, individuals exhibited a broad palette of colours (ranging from cream and light orange tonalities to dark green) regardless of the presence of oocytes in their coelom.

We found ovigerous females all throughout the study. Still, in November 2019, all the females had such a low density of oocytes that it was not reached the minimum sample (45 oocytes) necessary to calculate the mean oocyte size. We called these specimens “specimens with residual oocytes”, i.e., specimens at the end of the maturation process that had spawned most of the oocytes they originally contained, or specimens at the beginning of the process of generating new oocytes for their subsequent maturation. We also found other “specimens with residual oocytes” within the period September–February, even though the highest frequency occurred in both years in the month of November (13.1%; Table 1). The mean W_10_ of the specimens with oocytes was 9.72 ± 0.04. The W_10_ of the smallest and biggest specimens with oocytes was 7.18 mm and 11.05 mm, respectively.

Our data showed that both the percentage of individuals with oocytes and the mean oocyte size displayed an increasing trend from their annual minimum in November–January compared to the months of May–September (Figure 3, Figure 4 and Figure 5). In particular, the percentage of individuals with oocytes exhibited its lowest value in November 2019 (0%) and November 2020 (14.75%), while it was as high as 51.22% in July 2019 and 57.81% in September 2020 (Figure 4). Likewise, the mean oocyte size exhibited its lowest value in January 2020 (80.19 µm) and December 2020 (115.84 µm) (Figure 5), while the maximum mean oocyte size was recorded in September both in 2019 and 2020 (157.59 µm and 166.83 µm, respectively). We found oocytes of less than 60 µm throughout all sampled months, and the standard deviation of the oocyte size over the entire study period was 26.93 µm. On the other hand, while a clear increase in both the percentage of individuals with oocytes and the mean oocyte size was observed in December–January, once the maximum value was reached, there was no longer an evident pattern (Figure 4 and Figure 5).

Additionally, during both reproductive seasons, the maximum percentage of ovigerous females was followed by a sharp drop in the following month. Thus, in 2019, the maximum percentage of ovigerous females registered was in July (51.22%), followed by a drop to 25.00% in August; moreover, in 2020, the highest percentage of ovigerous females occurred in September (57.81%) and decreased to 25.8% in November. In 2019, interestingly, the pronounced decrease in the percentage of ovigerous specimens coincided with the continuous drop in water temperature from August–September (Figure 4). In 2020, the water temperature showed pronounced variations between June and October, increasing from 14.04 °C in July to 20.27 °C in August, followed by a further increase in the percentage of ovigerous females in September (from 18.33% in August to 38.98% in September).

The correlation analyses showed a significant positive correlation between the percentage of individuals with oocytes and the water temperature (*r* = 0.555; *p* < 0.05) as well as the photoperiod (*r* = 0.537; *p* < 0.05), while there was no correlation with salinity, primary production (whether UV fluorescence or fluorescence) and upwelling index (Table 3). A correlation was also significantly positive between the mean oocyte size and the water temperature (*r* = 0.445; *p* < 0.05). The correlation between the weight and the W_10_ was significant (*r* = 0.621; *p* < 0.05). On the other hand, no correlation was observed between mean oocyte size and individual size (estimated either by W_10_ or by individual weight) (*p* < 0.05; Table 3).

## 4. Discussion

*Diopatra neapolitana* is a marine resource of economic importance and should be managed for sustainability. The first key point for proper management is the correct identification of the species. Species that are morphologically very similar may not share the same biological characteristics, including their feeding mode, their habitat and depth preferences, their degree of tolerance to various environmental parameters and, also, their reproductive biology [39]. Therefore, cryptic or misidentified species can have implications for biodiversity conservation and management [40]. Although *Diopatra neapolitana* is the only species within this genus that has been recorded in Galicia up to date, this species has been found sympatrically with *D. marocensis* [12,41] and *D. biscayensis* [14] in other localities both southwards and northwards. The species *D. micrura*, also observed in areas geographically close to Galicia, usually occupies, however, the subtidal zone [7], and so it is more unlikely to occur together with *D. neapolitana* in our study area. The morphological analyses carried out over a representative subsample of the individuals collected in the present study confirmed that the *Diopatra* intertidal individuals from A Portela bank (Ría de Vigo) belong to the species *D. neapolitana*. Arias et al. (2016) [10] already collected three *Diopatra* specimens from a location very close to our study area (approximately 2 km away) that were molecularly identified as *D. neapolitana*. Likewise, in the framework of another study, we sequenced COI-5P from two specimens collected in this work; these sequences, which were identical, supported our morphological analysis and revealed that *Diopatra* specimens from A Portela bank are *D. neapolitana* (GenBank accession numbers: OP093375 and OP093376). These two individuals were deposited as DNA voucher specimens of *D. neapolitana* at the Biological Research Collection (Marine Invertebrates) of the Department of Biology of the University of Aveiro, Portugal (CoBI-DBUA) (DBUA0002489.07; DBUA0002489.08). In this regard, it is worth mentioning that the use of molecular techniques is gaining importance in the taxonomy field, as their use is increasingly revealing new identities in already known species; indeed, in the particular case of polychaetes, previously overlooked species have been discovered in the main families of the group [39].

As for the reproductive cycle of *D. neapolitana,* our results suggest that two reproductive cycles occurred during the studied period from May 2019 to April 2021. The cycles began annually in the colder months of January/February, when the observed oocytes were smaller on average (ranging from 80 to 124 µm) and matured to gain their largest size in September/October (ranging from 157 to 167 µm). Furthermore, no correlation was found between the mean oocyte size and the size of individuals, probably due to the narrow size range of the studied specimens. The mean standard deviation of the oocyte size was high all along this period, except for September/October when the oocytes reached their maximum size. In addition, we observed some oocytes of less than 60 µm in each and every one of the months of the study. These results suggest that individuals hold oocytes at different stages of maturation and, therefore, oogenesis takes place in *Diopatra* females asynchronously, which confirms the observations for *D. neapolitana* from Arias et al. (2016) [10] in the Villaviciosa estuary (northern Spain) and Bergamo et al. (2019) [11] in Araçá Bay, Southwestern Atlantic.

The spawning time of polychaete populations has been frequently inferred from the proportion of individuals with oocytes, assuming that a sharp decline in this proportion does not imply a decrease in the number of females, but rather that there are females that have spawned and no longer have oocytes [10,11,34]. The evolution of the percentage of ovigerous females in the present study suggests that spawning occurred asynchronously from July/August to November/December, when a resting period was reached and no more females with oocytes > 60 µm were observed. Thereafter, a new maturation cycle starts again. In the first studied period, the spawning began in August 2019 and the percentage of ovigerous females continued decreasing gradually until all observed females were empty in November/December. However, in the second year, this pattern was not so evident, as once the first spawning started in July, a partial recovery in the percentage of ovigerous females was observed in September 2020. Despite not having taken data on the fecundity of the analysed specimens, we observed a decrease in the number of oocytes in the months that proceeded the sudden drop in the percentage of specimens with oocytes in July 2020, supporting the theory that this sudden decrease corresponds to a spawning event in the population. However, it should be noted that the sharp decline in the percentage of females with oocytes observed in this study determines the “most massive” spawning event prior to the resting period, but it does not imply that other minor spawning events could be occurring continuously at times prior to or after this sharp decline.

The water temperature recorded during the summer of 2020 suggests that the sharp variations in this environmental variable were responsible for the observed spawning and subsequent partial recoveries. In particular, significant drops in water temperature seem to have triggered spawning (both in August 2019 and July 2020), while the outstanding temperature increase from 14 to 20 °C observed between July and August 2020 seems to have promoted the new maturation of oocytes, as reflected by the following increase in ovigerous females observed in September 2020. The role of water temperature in driving the induction of the spawning has been demonstrated in other polychaetes species such as *Ptychodera flava* [42], *Hediste diversicolor* [43] and *Arenicola loveni loveni* [44], in which the spawning timing can be induced in laboratory conditions.

Furthermore, the influence of the water temperature on the reproductive cycle of *D. neapolitana* is supported by the significant positive correlation between the superficial water temperature and both the proportion of individuals with oocytes and the mean oocyte size. The surprisingly significant drop in the water temperature in the warmer months that triggers spawning is due to the coastal upwelling of deep cold North Atlantic Central Water, a phenomenon that occurs frequently in Galicia, especially in the spring and summer, and which has been extensively studied [45,46]. Indeed, the upwelling index was 80% higher in July 2020 than in June and September 2020 (Table 1). These upwelling events have been shown to influence the reproductive cycle of various marine invertebrates in NW Spain, from copepods such as *Calanoides carinatus* and *Calanus helgolandicus* [47] to the common octopus *Octopus vulgaris* [48]. With all that is mentioned above, it is somewhat astounding that we did not find a significant correlation neither between the upwelling index and the mean oocyte size, nor between the upwelling index and the percentage of specimens with oocytes. This may be explained by the fact that even though the upwelling index and temperature are closely related, the former refers to the movement of deep (cooler) water masses that rise to the surface, even if the cooling effect of surface water does not always occur immediately after upwelling. In fact, these two variables are significantly correlated (*r* = 0.526; *p* < 0.05) when applying a 1-month lag. On the other hand, the rise in water temperature in August 2020 after the end of the July upwelling event was followed by an increase in the proportion of individuals with oocytes in September and October 2020. However, once again this increase did not occur immediately after the rise in water temperature, but with a month’s delay, which may be due to a “lag” effect, since the consequences of environmental changes on the biology of an individual are not immediate but need some time to manifest themselves [49,50].

Variations in water temperature might explain the differences reported in the reproductive cycle of *D. neapolitana* at sites with different annual water temperature dynamics. For example, in Ria de Aveiro (N-Portugal) and Ria de Villaviciosa (Cantabrian coast of Spain), where mean water temperature also varies seasonally throughout the year, *D. neapolitana* showed a discontinuous reproductive season with a main single annual spawning event [10,34], whereas on the tropical coast of São Sebastião (Brazil), with a constant average water temperature throughout the year, *D. neapolitana* showed a continuous reproductive cycle [11]. The reproductive cycle being discontinuous in temperate regions and continuous in tropical waters has also been observed in other polychaete species, such as *Streblospio shrubsolii* [51] and *Streblospio gynobranchiata* [52]. Furthermore, even in the case of continuous reproduction throughout the year, or the fact that an extended breeding period is common for marine invertebrates in tropical regions [53], it is important to note that the Brazilian population of *D. neapolitana* described by Bergamo et al. (2019) [11] is outside its natural range of distribution as it is an introduced species. The timing of its reproductive cycle may therefore not only be influenced by environmental conditions but is a plastic response that helps maximise the reproductive success in the introduced species [54,55]. Differences in reproductive biology between native and non-native localities have been studied for other marine invertebrate species, with clear results in this regard [56,57]. However, in many other cases, it was difficult to determine whether differences in the timing of the reproductive cycle were due to differences in the environmental conditions related to the place of origin, or whether they were related to the flexibility of these species. In any case, the importance of temperature and photoperiod in the reproductive cycle of *D. neapolitana* agrees with what has been concluded for other polychaete species, for which both photoperiod and temperature have been observed to be determinant factors in gametogenesis, as well as spawning timing [58,59,60,61].

Rising temperatures due to climate change have been shown to have a significant effect on the biology of some marine invertebrate species. Specifically, temperature plays an important role in the reproductive cycle of *D. neapolitana*. In spite of our results, further studies are needed to know exactly how temperature affects the reproductive physiology and, indirectly, the abundance and distribution of this and other similar species.

## Figures and Tables

**Figure 1 biology-11-01504-f001:**
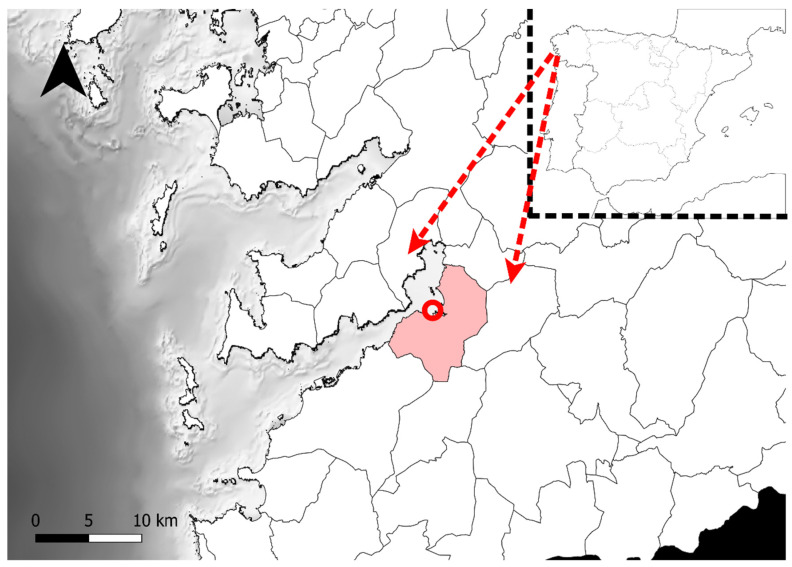
Map of Ría de Vigo with Redondela highlighted in pink. The shellfish bank A Portela (VI-132), where the *Diopatra* specimens were collected, is marked with a red circle (42°17′25.2″ N, 8°37′24.4″ W).

**Figure 2 biology-11-01504-f002:**
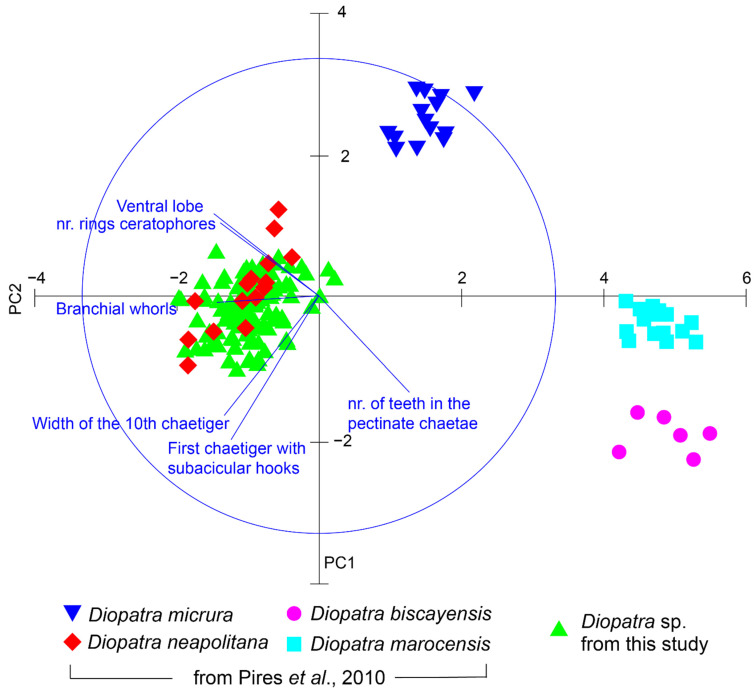
Results of the ordination after the application of Principal Components Analysis to the main morphological descriptors (width of the 10th chaetiger, number of rings in the ceratophores, maximum number of branchiae whorls, first chaetiger with subacicular hooks, number of teeth in the pectinate chaetae and presence/absence of a ventral lobe in the parapodia) of 105 specimens of *Diopatra* sp. from our study (Redondela, NW Spain), compared with the specimens studied in Pires et al. (2010) [11]. Each point represents one individual. Our specimens form and isolated cluster with those of *Diopatra neapolitana* from Pires et al. (2010) [11], confirming their identification as *Diopatra neapolitana*.

**Figure 3 biology-11-01504-f003:**
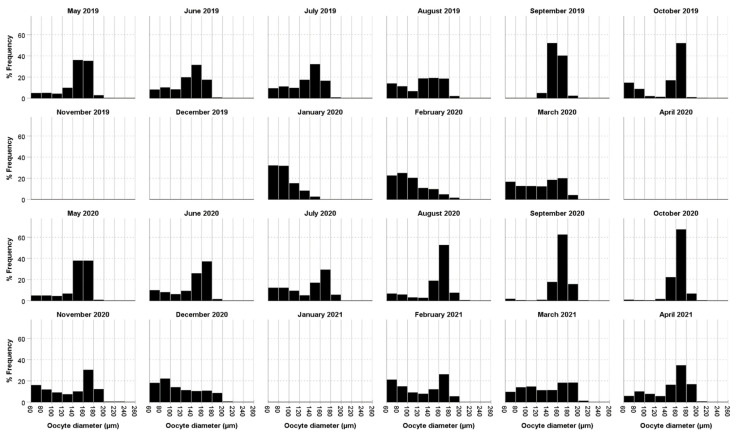
Monthly distribution of oocyte sizes of *Diopatra neapolitana* from Redondela (NW Spain). No samples were caught in the months of December 2019, May 2020 and January 2021.

**Figure 4 biology-11-01504-f004:**
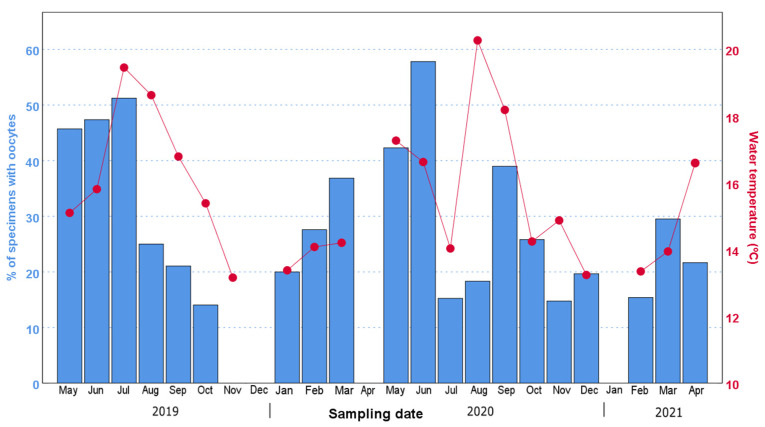
Temporal variation in the percentage of specimens with oocytes (bars) and the water temperature (dots and lines) over the 2-year period studied in this work. The months without data are due to problems derived from COVID-19, or to months in which the weather did not allow us to collect samples.

**Figure 5 biology-11-01504-f005:**
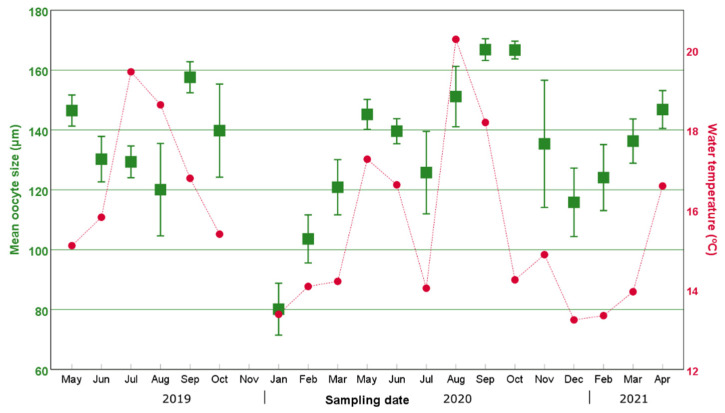
Temporal variation in the mean oocyte size (squares) over the 2-year period studied in this work and the water temperature (dots and lines). Error bars correspond to the standard deviation of the mean oocyte size of the specimens.

**Table 1 biology-11-01504-t001:** Summary of the reproductive variables analysed for those specimens of *Diopatra neapolitana* showing oocytes in the coelom, as well as the environmental variables collected for the study area. Presence of oocytes < 60 µm is shown as a qualitative indicator of the presence of oocytes at the beginning of the maturation stage. Residual oocytes refer to specimens that had oocytes but whose number was so low that their mean size could not be calculated. * The weight was measured only in complete specimens.

				Oocyte Analysis					Environmental Variables					
Data	N	W10 *	Weight	Specimens with Oocytes (%)	Mean Oocyte Size (µm)	Standard Deviation	Presence of Oocytes < 60 µm	Residual Oocytes	Water Temperature (°C)	Photoperiod	Salinity	UV Fluorescence (µg/L)	Fluorescence (mg/m^3^)	Upwelling Index (m3 s^−1^ km^−1^)
May-19	35	8.56		45.71	146.47	9.36	Yes	No	15.10	0.61	34.88	4.65	−0.24	362,989
Jun-19	38	9.49		47.37	130.24	15.18	Yes	No	15.81	0.63	35.19	4.52	−0.22	−16,593
Jul-19	41	9.54	10.49	51.22	129.34	11.87	Yes	No	19.46	0.61	34.82	0.28	−0.44	598,945
Ago-19	40	9.98	11.16	25	120.05	24.41	Yes	No	18.83	0.55	35.41	0.24	−0.44	200,653
Sep-19	38	9.71	11.89	21.05	157.59	7.33	Yes	Yes	16.79	0.50	35.34	0.27	−0.44	700,942
Oct-19	64	9.64	11.06	14.06	139.78	23.33	Yes	Yes	15.39	0.43	33.74	0.28	−0.44	−45,088
Nov-19	61			0			Yes	Yes	13.16	0.39	28.96	0.33	−0.44	162,748
Jan-20	60	9.85	13.54	20	80.13	16.8	Yes	Yes	13.38	0.39	33.53	0.27	−0.44	−411,905
Feb-20	58	9.97	12,37	27.59	103.60	16.05	Yes	Yes	14.08	0.44	31.50	0.29	−0.44	−68,465
Mar-20	57	9.73	12,24	36.84	120.87	20.61	Yes	No	14.21	0.49	31.1	3.48	0.06	374,427
May-20	26	9.58		42.31	145.19	12.19	Yes	No	17.27	0.6	29.29	3.98	−0.01	468,078
Jun-20	69	9.63	11.09	15.25	139.59	20.66	Yes	No	16.63	0.63	34.60	3.94	−0.23	196,981
Jul-20	60	9.57	11.59	18.33	125.75	15.95	Yes	No	14.04	0.62	36.13	4.97	−0.16	1,801,995
Ago-20	59	9.71	10.69	38.98	151.16	8.72	Yes	No	20.27	0.57	34.75	−0.03	−1.22	−45,013
Sep-20	64	9.9	13.65	57.81	166.83	8.29	Yes	Yes	18.19	0.51	35.02	−0.03	0.85	522,912
Oct-20	62	9.9	14.04	25.81	166.69	5.90	Yes	Yes	14.24	0.45	35.20	−0.03	1.79	−205,039
Nov-20	61	10.35	16.15	14.75	135.36	31.85	Yes	No	14.88	0.4	32.65	−0.03	1.79	−280,501
Dec-20	61	10.23	14.64	19.67	115.84	19.81	Yes	Yes	13.24	0.38	21.69	−0.03	2.96	−96,067
Feb-21	65	10	14.44	15.38	124.09	17.42	Yes	Yes	13.35	0.42	12.3	−0.03	3.39	−2,073,219
Mar-21	61	9.47	10.89	29.51	136.28	11.41	Yes	Yes	13.95	0.47	31.92	−0.03	19.89	619,616
Abr-21	60	9.88	14.48	21.67	146.82	15.7	Yes	No	16.60	0.58	33.12	5.32	4.77	413,329
Total	1130			28.01	134.5	26.93								

**Table 2 biology-11-01504-t002:** Summary of the six diagnostic morphological characters. Average (total number of chaetigers, width of the 10th chaetiger) and range (number of rings on the ceratophores, maximum number of branchial whorls, first chaetiger with subacicular hooks and number of teeth in pectinate chaetae) values obtained in the 105 Diopatra specimens collected in Redondela (NW Spain) during the present work as well as the D. neapolitana specimens from Rodrigues et al. (2009) [12] (*n* = 15). The total number of chaetigers of complete specimens is also shown; n/d: not determined.

Character	*Diopatra neapolitana* Redondela (*n* = 105)	*Diopatra neapolitana* from Rodrigues et al. (2009) (*n* = 15)
Total number of chaetigers (complete specimens)	257.02 ± 3.17 (*n* = 48)	n/d
Width of the 10th chaetiger (mm)	9.25 ± 0.07	7.2 ± 0.42
Number of rings on the ceratophores	12–18	15–16
Maximum number of branchial whorls	12–20	14–18
First chaetiger with subacicular hooks	17–24	19–25
Number of teeth in pectinate chaetae	5–9	5–10
Presence of a ventral lobe in the parapodia	Present	Present

**Table 3 biology-11-01504-t003:** Pearson’s (r) or Kendall’s (*τ)* coefficients from correlation tests between the reproductive parameters (mean oocyte size and percentage of individuals with oocytes) and the six environmental variables gathered for the study area, as well as the biometric data taken for each individual. ^a^ Kendall rank correlation tests were used in these variables. *p* value (P) is also shown.

		Water Temperature	Photoperiod	Salinity	UV Fluorescence	Fluorescence	Upwelling Index	W_10_	Weight
**Mean Oocyte Size**	r/τ ^a^	0.445	0.34	0.242 ^a^	0 ^a^	0.064 ^a^	0.228	−0.027	0.114
	*p*	0.049	0.143	0.136	1	0.696	0.335	0.683	0.109
**Percentage of Individuals with Oocytes**	r/τ ^a^	0.555	0.537 ^a^	0.162 ^a^	0.059 ^a^	−0.034 ^a^	0.237		
	*p*	0.009	0.012	0.305	0.713	0.832	0.3		

## Data Availability

Data are available in a publicly accessible repository: https://doi.org/10.5281/zenodo.7081249 (accessed on 14 September 2022).

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
