# Peer review of "Insights into Environmental Drivers on the Reproductive Cycle of Diopatra neapolitana (Polychaeta: Onuphidae)"

_biology, 2022, doi:10.3390/biology11101504_

Round 1

Reviewer 1 Report

Review

Paper title: Insights into Environmental Drivers on the Reproductive Cycle of Diopatra neapolitana (Polychaeta: Onuphidae)

The authors conducted a long-term study to describe the most important aspects of the reproductive biology of the polychaete Diopatra neapolitana in NW Spain. The authors presented new data concerning the presence of oocytes and their size. They also provided morphological characteristics of the worms studied that may be useful for morphological identification of sympatric species. The authors concluded that Diopatra neapolitana exhibits a discontinuous reproductive cycle, with proliferation and maturation of oocytes from January to September and asynchronous spawning events between July and October. A resting period occurs in November and December. The authors found significant correlations of reproductive parameters with water temperature and concluded that this factor is the most important driver of reproductive events in Diopatra neapolitana.

All these reasons explain the relevance of the paper by Daniel Escobar-Ortega and co-authors submitted to "Biology".

General scores.

The data presented by the authors are original and significant. The study is correctly designed and the authors used appropriate sampling methods. In general, statistical analyses are performed with good technical standards but additional analyses are required. The authors conducted careful work that may attract the attention of a wide range of specialists focused on benthic ecology.

Suggestions.

L 72-72. The aim of this study should be placed at the end of the Introduction.

The authors used a parametric test (Pearson correlation analysis). This approach requires normal distribution and heterogeneity of the data. The authors should provide the methods used to test these assumptions.

L 220-232. The authors should compare the data among different periods using statistical methods such as chi-square tests (percentage data) and ANOVA (mean oocyte diameter).

L 246. The authors stated that "The Spearman correlation analyses…". Please, clarify which analysis was used in the study.

Table 1. What do the authors mean by "Mean standard deviation"? Please specify how it was obtained.

The authors should use the full Latin names on figures, tables and legends.

L 228. What do the authors mean by "the average standard deviation"? Please specify how it was obtained.

Specific remarks.

L 26. Consider replacing “polychaeta” with “polychaete”

L 27. Consider replacing “characterized for” with “characterized by”

L 38. Consider replacing “in the Araçá” with “in Araçá”

L 39. Consider replacing “described in” with “described along”

L 42. Consider replacing “recorded in” with “recorded on”

L 43. Consider replacing “registered in” with “registered on”

L 53. Consider replacing “the large number” with “a large number”

L 55. Consider replacing “knowledge on” with “knowledge of”

L 56. Consider replacing “increasingly being” with “increasingly”

L 61. Consider replacing “in the Izmir” with “in Izmir”

L 61. Consider replacing “in the Araçá” with “in Araçá”

L 64. Consider replacing “larvae” with “larva”

L 72. Consider replacing “biology of the” with “biology of”

L 76. Provide a number citation for Pires et al. 2010. The same is relevant for other citations in the text (L 111, 115, 124, 148, 178

L 77. Consider replacing “may have a very” with “may have very”

L 101. Consider replacing “moment” with “the moment”

L 116. Please, clarify "that six seven characters discriminates": "six to seven" or "six of the seven"? Change "discriminates" to "discriminate"

L 124. Consider replacing “to a representative” with “into a representative”

L 134. Consider replacing “oocytes of” with “oocytes”

L 186. Consider replacing “grouped in” with “grouped at”

L 196. Consider replacing “is also show” with “is also shown”

L 204. “D. neapolitana” should be italicized.

L 207. Consider replacing “observed” with “observe”

L 208. Consider replacing “immatures” with “immature”

L 227. Consider replacing “in 2019 and in 2020” with “2019 and 2020”

L 231. Consider replacing “observed from” with “observed in”

L 234. Consider replacing “was followed” with “followed”

L 235. Consider replacing “was followed” with “followed”

L 249. Consider replacing “Correlation” with “A correlation”

L 257. Consider replacing “evolution of” with “variation in”

L 262. Consider replacing “evolution of” with “variation in”

L 263. Consider replacing “corresponds with” with “correspond to”

L 279. Consider replacing “collected on” with “collected in”

L 298. Consider replacing “excepting the months of” with “except”

L 318. Consider replacing “of July” with “in July”

L 326. Consider replacing “The water temperature being a driver factor  in the induction of the spawning” with “The role of water temperature in driving the induction of the spawning”

L 353. Consider replacing “The temperature being a main driver of the reproductive cycle on this 

species,” with “Variations in water temperature”

L 354. Consider replacing “on the reproductive cycle” with “in the reproductive cycle”

L 363. Consider replacing “a continuous reproduction” with “continuous reproduction”

L 368. Consider replacing “but being” with “but is”

Reviewer 2 Report

Insights into Environmental Drivers on the Reproductive Cycle of Diopatra neapolitana (Polychaeta: Onuphidae)

Review

The purpose of the present study was “to advance in the knowledge of the reproductive biology of the Diopatra neapolitana”, as it implicitly stated on line 72. To achieve the aim, authors examined coelomic oocytes which they obtained by filtering coelomic fluid, extracted from ethanol-fixed big females, through a 60-µm and 500-µm sieves. Comparing monthly measurements of the oocytes with some relevant environmental variables, they showed that “in the Ría de Vigo, the reproductive cycle seems to be influenced by upwelling events of deep cold water that occur frequently between March and September, and that could act as a spawning-inducing switch.” This, actually, is the major conclusion inferred from this study.

Another background purpose of the present study was to gain knowledge of the reproductive biology of Diopatra neapolitana “as an essential component of effective fisheries management” (Lines 50-53). Such management, however, first of all requires knowledge of the maturation age, longevity, fecundity, periods and number of spawning events during the year and the whole lifespan. Very little of this was discovered during this study.

A number of particular comments I provided directly in the body of the manuscript. My major comments are shown below:

Lines 72-85. Last paragraph of the Introduction.

Re:  I strongly recommend to provide here more available information about Diopatra neapolitana, first of all information about reproductive biology of the species obtained by earlier authors. Protandrous hermaphroditism and size of maturation reported by Arias et al. (2016) for the population from northern Spain! would help to better understand new data obtained by the authors of the submitted manuscript.

Lines 77-78: “similar species may have a very different reproductive biology

Re: Indeed, Onuphidae is a very interesting family which members demonstrate diverse and some unique kinds of oogenesis and larval development. For some reason, the authors mention only larval development patters distinguished by Paxton (1993) and say no a word about oogenesis, although their study deals with this aspect.

Lines 106-107: “We also measured the width of the 10th chaetiger (W10), a metric employed as a proxy of body size in Diopatra studies

Re: The total number of chaetigers is well known and reported for many polychaetes as a major variant that is strongly correlated with many other individual (morphologic, reproductive, etc.) characteristics. Why only width of the 10th chaetiger is used for Diopatra? If because most sampled worms are incomplete, it should be clearly stated.

Lines 108-109: For examination, “whole animals were individually preserved in ethanol 70%”.

Re:

Lines 130-134: “This coelomic solution was first filtered through a 60 μm sieve and the particles retained were next filtered through a 500 μm sieve

Re: The technique used to examine the reproductive cycle seems to me rather inappropriate. The authors got live worms in the laboratory! Thus, all the procedures could be done on relaxed (anesthetized) live individuals. Dissecting live worms, the authors could get and examine in vivo not only ALL the coelomic oocytes, but also the oocytes developing inside ovaries (early stages of oogenesis occur in ovaries). 70% ethanol is inappropriate fixative for polychaetes, what is mentioned in every manual for polychaete examination. Ethanol causes agglutination of coelomic elements and makes work with them difficult. Filtering coelomic liquid first through 60 µm and then 500 µm sieves certainly results in loosing of a great part of the coelomic elements, including oocytes. Formalin would be much more appropriate fixative than ethanol. But, to avoid use of this dangerous chemical, working on live material would be the best and the most appropriate for this study. Various techniques of oocyte examination were described in details by Anderson, E. & Huebner, E. (1968) Development of the oocyte and its accessory cells of the polychaete, Diopatra cuprea (Bosc). Journal of Morphology, 126 (2), 163-172.

Precise examination of anesthetized live adults and their coelomic fluid was done by Arias, A., Paxton, H. & Budaeva, N. (2016) Redescription and biology of Diopatra neapolitana (Annelida: Onuphidae), a protandric hermaphrodite with external spermaducal papillae. Estuarine, Coastal and Shelf Science, 174, 1-17.

Line 134: We also recorded the presence of oocytes of smaller than 60 μm.

Re: What was the smallest diameter of oocytes to consider a female ovigerous?

Authors bring unneeded confusion about the number of morphological characters examined for the identification of their specimens. On Line 118 (Section 2.3) they wrote: “We studied these six characters”, whereas on line 146 they state: “Morphological data in relation to the seven diagnostic characters mentioned in section 2.3”.

Lines 156-157: “The maturation cycle was studied through the monthly variation of the mean oocyte size.

Re: What do you call “maturation cycle”? Without explanation, the term sounds confusing. Moreover, if new oocytes develop continuously and individuals have several spawning events during a reproductive season, it is impossible to describe maturation cycle and both individual and population by simple measurements of the coelomic oocytes within the whole population.

Lines 157-158: “spawning events were identified by sharp drops in the percentage of individuals with oocytes

Re: This way identifies only last spawning event within the population before it goes for a resting period. But possible spawning events remain unrecognized if females produce oocytes continuously and have several asynchronous events during an extended reproductive period.

Lines 159-160: “We used Pearson correlations to explore the relationship between the reproductive descriptors (i.e. mean oocyte size and the percentage of individuals with oocytes)

Re: This is a confusing and possibly misleading approach. 1) For the “individuals with oocytes” the minimum diameter of the oocytes must be specified. 2) “individuals without oocytes” can include two categories: a) yet immature individuals, and b) worms just after spawning.

Line 174: 3.1. Morphological identification

Re: The way to identify Diopatra worms used by the authors seems unusual to me, but, if this is a widely accepted approach, it’s okay. My only concern is that individuals of different sizes were compared for species diagnostic – see Table 2. The authors should say something about this, because many polychaetes are known to change morphological characteristics during growth.

Lines 206-219: 3.2. Reproductive cycle

Re: The observations obtained by the authors are interesting, but partly meaningless or incomplete if not associated with size of the examined individuals. It is rather strange that the individual size is not discussed in this section at all, although authors mention absence of correlation with the body colour and the presence of oocytes. What about correlation of individual size and the presence of oocytes? Only 26% of the examined worms had oocytes. How do you explain the other 74%? All this section raises too many questions and does not provide answers…

Lines 213-216: “"specimens with residual oocytes", i.e., specimens at the end of the maturation process that had spawned most of the oocytes they originally contained, or specimens at the beginning of the process of generating new oocytes for their subsequent maturation.

Re: Same problem again: size of the examined individuals could shed some light and provide explanation on the obtained information. Without this parameter involved into consideration, many questions remain without answer.

Lines 253-254: “no correlation was observed between mean oocyte size and individual size (estimated either by W10 or by individual weight)

Re: This statement, as it is, clearly shows incorrect methodology used by the authors but not the reproductive characteristics of worms. First of all, the size of maturation should be determined for the examined species. And only after that reproductive characteristics should be examined in mature adults only. Without consideration of total size and maturity size, all the obtained data appear confusing and at least some conclusions sound fairly strange.

This very misleading statement is repeated on lines 296-297.

I can say that the authors insufficiently show and refer to the information provided by other authors. For example, Arias et al. (2016) reported protandrous hermaphroditism in D. neapolitana and showed that smallest ovigerous specimen was 118 mm long and 3.8 mm wide. From these data provided by Arias et al. (2016), I can see that the authors of the revised manuscript dealt with mature females only.

Lines 281-288: we sequenced COI-5P from two specimens collected in this work; these sequences, that were identical, supported our morphological analysis and revealed that Diopatra specimens from A Portela bank are D. neapolitana... In this regard, it is worth mentioning that the use of molecular techniques is gaining importance in the taxonomy field”

Re: It would also be worth mentioning that Arias et al. (2016) already sequenced some Diopatra from Ría de Vigo and referred them to D. neapolitana.

Lines 299-302: “In addition, we observed some oocytes less than 60 μm in each and every one of the months of the study. These results suggest that individuals hold oocytes at different stages of maturation and, therefore, oogenesis takes place in Diopatra females asynchronously,

Re: Yes, indeed, there is no synchronization of the oogenesis within the population. But what was much more important is to show if the oogenesis was continuous, when females steadily produce oocytes.

Well, I can say that got answers to many of my questions reading Arias et al. (2016). Monthly size-frequency histograms of coelomic oocyte diameter of D. neapolitana provide on Fig. 7 by those authors is very informative. Showing similar histograms for the D. neapolitana population from Redondela, would answer many questions which come to readers after reading the present version of the manuscript.

Supplementary Figure 1. Nice SEM images. Pity that they do not show (and not discussed in the text at all!) spermaducal papillae first described for the species by Arias et al. (2016).

Round 2

Reviewer 1 Report

The authors have revised the paper accroding to my comments.

Reviewer 2 Report

I am glad that the authors improved the text according to reviewers' comments